# MM-Ego: Towards Building Egocentric Multimodal LLMs for Video QA

🍎**Hanrong Ye**[1][†]**, Haotian Zhang**[2][†]**, Erik Daxberger**[2]**, Lin Chen**[2]**,** 🍎**Zongyu Lin**[3]**, Yanghao Li**[2]**, Bowen Zhang**[2]**, Haoxuan You**[2]**, Dan Xu**[1]**, Zhe Gan**[2][‡]**, Jiasen Lu**[2][‡]**, Yinfei Yang**[2][‡]

[1]CSE, HKUST    [2]Apple    [3]UCLA
hanrong.ye@connect.ust.hk, {haotian.zhang2,yinfeiy}@apple.com

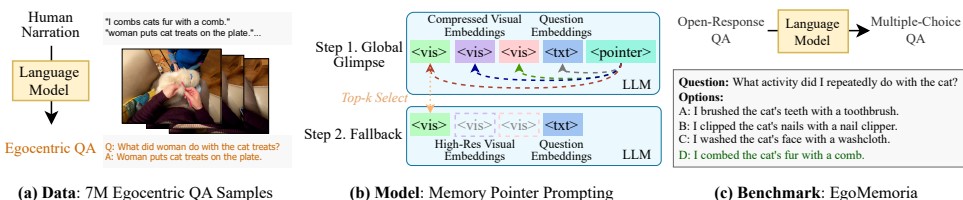

(a) **Data**: 7M Egocentric QA Samples    (b) **Model**: Memory Pointer Prompting    (c) **Benchmark**: EgoMemoria

Figure 1: We introduce a foundation model for egocentric video understanding, contributing from three key perspectives: (a) 7 million egocentric QA samples generated from human narrations via a data engine, (b) a multimodal language model designed for egocentric video comprehension, and (c) the curation of a challenging egocentric video understanding benchmark.

## Abstract

This research aims to comprehensively explore building a multimodal foundation model for egocentric video understanding. To achieve this goal, we work on three fronts. First, as there is a lack of QA data for egocentric video understanding, we automatically generate 7M high-quality QA samples for egocentric videos ranging from 30 seconds to one hour long in Ego4D (Grauman et al., 2022) based on human-annotated data. This is one of the largest egocentric QA datasets. Second, we contribute a challenging egocentric QA benchmark with 629 videos and 7,026 questions to evaluate the models' ability in recognizing and memorizing visual details across videos of varying lengths. We introduce a new de-biasing evaluation method to help mitigate the unavoidable language bias present in the models being evaluated. Third, we propose a specialized multimodal architecture featuring a novel "Memory Pointer Prompting" mechanism. This design includes a *global glimpse* step to gain an overarching understanding of the entire video and identify key visual information, followed by a *fallback* step that utilizes the key visual information to generate responses. This enables the model to more effectively comprehend extended video content. With the data, benchmark, and model, we build MM-Ego, an egocentric multimodal LLM that shows powerful performance on egocentric video understanding.

## 1 Introduction

Study on egocentric videos explores how machines can see and understand the world from a first-person, self-centered perspective. Egocentric videos differ significantly from static-camera videos, such as movies or animations, both in terms of content and viewpoint. The content of egocentric videos primarily revolves around human daily activities. These videos typically share a perspective similar to human vision, where the camera and viewpoint frequently move. As a result of these characteristics, egocentric videos exhibit a distinct data distribution compared to static-camera videos,

---

🍎Work done during an internship at Apple. [†]First Authors. [‡]Senior Authors.

which has motivated a new area of research. In recent years, research interest in egocentric intelligence has been on the rise (Sigurdsson et al., 2018; Damen et al., 2018; Grauman et al., 2022; Mangalam et al., 2023; Plizzari et al., 2024). This growing interest is driven by the rapid advancements in AR/VR headsets and robotics, where cameras capture long-form egocentric videos in a manner akin to human vision. Research on egocentric videos will allow these devices to understand their surroundings and human intentions, fostering more advanced machine intelligence and improving the human-machine interaction experience, with immeasurable research and application potential.

However, research on understanding egocentric videos remains in its early stages, with previous research primarily centered on specialized tasks such as story summarization (Lee et al., 2012), hand-object relationship understanding (Cai et al., 2016), action classification (Cartas et al., 2017; Li et al., 2021), and temporal or spatial grounding (Grauman et al., 2022). In contrast, works focusing on developing a more general egocentric video understanding model capable of complex understanding remain rare. Despite that video multimodal large language models (MLLMs) demonstrate strong video understanding and reasoning ability (Zhang et al., 2023a; Wang et al., 2024b; Lin et al., 2024; Zhang et al., 2024b), most of these works are unsuitable for egocentric video understanding from data, benchmark, and model design perspectives.

($a$) From a data standpoint, although many MLLMs use some egocentric videos from ActivityNet (Yu et al., 2019), Ego4D (Grauman et al., 2022), and Charades (Sigurdsson et al., 2018) in their training, they have not been trained on *large-scale* egocentric video datasets, which inherently restricts their ability to comprehend lengthy first-person videos and accurately extract visual details. While Ego4D (Grauman et al., 2022) offers valuable human-annotated videos and labels for certain egocentric video understanding tasks, particularly episodic memory (which assesses a model's ability to retain visual details in such videos), its annotations are not structured for generating language responses, making them unsuitable for training MLLMs. Therefore, a large-scale egocentric video QA corpus is still needed. ($b$) In terms of benchmarking, exisiting video QA benchmarks either focus on shorter videos – such as EgoSchema (Mangalam et al., 2023) and QaEgo4D, which evaluate using around 3-minute and 8-minute videos, respectively – or concentrate on Internet video content (e.g., Video-MME (Fu et al., 2024)). This creates a notable gap in egocentric video understanding benchmarks that encompass videos ranging from seconds to an hour in length. ($c$) From a model design perspective, previous video MLLMs have primarily addressed long videos in two ways. The first approach involves uniformly sampling a limited number of video frames as visual input, as seen in Li et al. (2024a); Lin et al. (2024). Despite its simplicity, this approach achieves better performance among open-source models on public video benchmarks (Fu et al., 2024), largely because its design ensures high training efficiency and good scaling properties. The second approach involves feeding a large volume of visual tokens into the transformer backbone and employing engineering techniques, such as tensor parallelism and sequence parallelism (Xue et al., 2024; Zhang et al., 2024a), to facilitate training with millions of visual tokens in context. However, these long-context transformers suffer from slow training speeds and small overall batch sizes, which hinder performance improvements given the constraints of computational resources and training time. Intuitively, even humans cannot remember every detail of an hour-long video. We believe a more effective approach is to understand the video progressively: first get an overview of the entire video, then focus on specific details with particular questions in mind.

Building on the observations mentioned above, we introduce MM-Ego, an egocentric MLLM designed to process and understand long egocentric videos. Our contributions are threefold:

($i$) **Data.** To scale training data for MLLMs with egocentric understanding ability, we develop an efficient data engine, using a "narration to egocentric QA" strategy, to automatically synthesize a large-scale egocentric QA dataset based on video narration data. Notably, rather than relying on existing vision-language models (VLMs) as labelers, we generate egocentric QAs based on the human-annotated fine-grained video clip narrations. This approach, conceptually related to (Di & Xie, 2024; Li et al., 2024b), ensures that our data quality is not constrained by the limitations of existing labeling VLMs. In this way, we create one of the first large-scale egocentric QA datasets, consisting of over 7 million egocentric QA samples that span video lengths from seconds to over an hour. This dataset enables the training of models to recognize and retain visual details from egocentric videos.

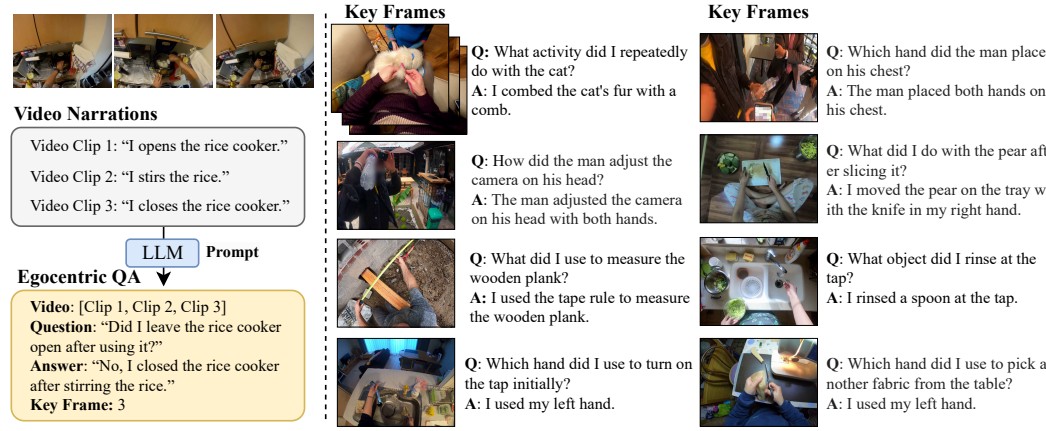

Figure 2: "Narration to Egocentric QA" data engine. Given a sequence of human-annotated video narrations, we instruct a language model (GPT-4o) to generate egocentric understanding-related questions and answers, along with identifying the key frames necessary to answer those questions.

(*ii*) **Benchmark.** To evaluate the MLLMs' performance in understanding and memorizing visual details from egocentric videos, we propose the EgoMemoria benchmark. This challenging benchmark includes 7,026 multiple-choice questions for 629 egocentric videos ranging from 30 seconds to 1 hour. In the experiments on EgoMemoria, we further investigate the impact of inevitable language biases across different models during evaluation and introduce a debiased metric to more accurately assess the models' true egocentric understanding capabilities.

(*iii*) **Model.** For our MM-Ego model, we develop a progressive approach to handle egocentric videos by introducing a Memory Pointer Prompting method. It consists of two steps: "global glimpse" and "fallback". In the *global glimpse* step, we extract compressed frame-level visual embeddings from the entire video to get a global understanding. Then, we employ a memory pointer embedding, designed to examine all compressed frame-level visual embeddings along with the question embeddings, to aid in identifying key visual embeddings in a question-aware manner. In the following *fallback* step, the selected key visual embeddings, in a higher-resolution form, are then used as final input to the LLM for processing and generation. This approach allows us to achieve a global understanding of the entire video while also identifying and utilizing key visual information to answer questions related to visual details.

## 2 METHOD

### 2.1 "NARRATION TO EGOCENTRIC QA" DATA ENGINE

As outlined in Section 1, high-quality egocentric QA pairs are lacking for training an MLLM with egocentric video understanding ability. To address this gap, we develop an innovative "narration to egocentric QA" data engine that automatically generates episodic memory-related QA samples based on human-annotated video clip narrations from the Ego4D dataset (Grauman et al., 2022) without the need for additional manual annotations.

Our approach leverages over 3,000 hours of privacy-protected, de-identified egocentric videos accompanied by more than 3 million high-quality, human-created narrations. These fine-grained language descriptions provide a rich resource for generating QA pairs.

The workflow of the data engine is illustrated in Figure 2. By organizing sequential video clips {Clip 1, Clip 2, ..., Clip N} and their corresponding narrations {Narration 1, Narration 2, ..., Narration N} in proper chronological order, we create comprehensive narration paragraphs that describe entire video sequences. We then employ a powerful **text-only language model**, i.e., GPT-4o, to generate diverse and confident QA pairs related to episodic memory based on these

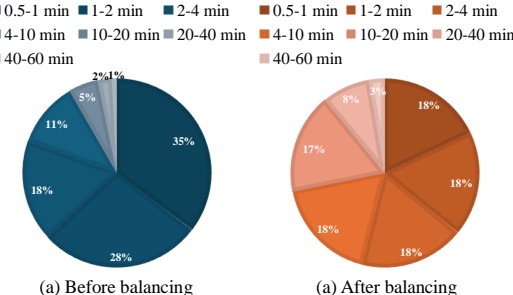

Figure 3: Video length distribution in our egocentric QA dataset.

narration paragraphs. The language model is instructed to attach the index of the narration sentence upon which each QA pair is based. This indexing allows us to map each QA pair back to the corresponding time frames in the original videos, enabling the extraction of key frame information crucial for subsequent model training.

Applying this data engine to the extensive Ego4D dataset allows us to efficiently scale the creation of egocentric QA data. We partition the dataset into training and testing sets according to the official Ego4D episodic memory task. The egocentric QA dataset provides more than 7 million QA samples in 938K multi-turn conversations. The data encompasses videos of varying durations, ranging from 30 seconds to 1 hour, as illustrated in Figure 3. To ensure comprehensive coverage and prevent bias towards shorter videos, we balance the number of conversations across different video lengths in training. This is one of the first large-scale egocentric QA datasets featuring videos of such extended ranges of duration.

Through these steps, our "narration to egocentric QA" data engine addresses the scarcity of large-scale, high-quality egocentric QA data for egocentric scenes, and sets a solid foundation for building MM-Ego, a sophisticated egocentric MLLM, which we introduce in the following section.

## 2.2 MM-Ego Model

Our modeling goal is to develop an MLLM for handling egocentric videos, which are lengthy and rich in visual details. On the one hand, frame-level information is necessary to capture the full content of the video, as skipping frames during sampling could result in a significant loss of visual details. On the other hand, processing all visual tokens generated by the visual encoder is computationally challenging for the transformer model. For instance, if each image is encoded into 729 visual embeddings (tokens), the total number of visual embeddings for a 300-frame video would be 218,700. However, most MLLMs are trained with a context length of less than 10,000 tokens (Li et al., 2024a). Taking these factors into account, we introduce the MM-Ego model, which is built for handling a large volume of egocentric video frames while maintaining manageable computational costs within the transformer backbone. MM-Ego introduces an innovative Memory Pointer Prompting mechanism, which operates in two main steps: global glimpse and fallback. We will introduce the details of MM-Ego in the following sections.

### 2.2.1 Visual and Textual Embedding

Given an input video and the question, the first step is to embed them into visual and textual embeddings separately for later processing. We begin by uniformly sampling the video into up to $N$ frames, where $N$ can be in the range of hundreds. Then, we extract per-frame visual feature maps from these frames using a robust vision encoder, SigLIP-so400m (Zhai et al., 2023). Following the method outlined by Li et al. (2024a), we apply a 2-layer MLP to project the visual feature maps to the LLM embedding space and use average pooling to reduce the height and width of the visual feature maps by a factor of two and flatten the height and width dimension, resulting in $N$ relatively high-resolution visual embeddings $\{\mathbf{V}^i \in \mathbb{R}^{T \times C}, i \in [1, N]\}$ where $T$ is the embedding length and $C$ is the embedding dimension. For the textual embedding, since we use Qwen2 (Yang et al., 2024) as the LLM, we use its tokenizer and embedding layer to transform the input text into textual embeddings. For question $q$, we denote the corresponding textual question embedding as $\{\mathbf{E}^q_{\text{que}} \in \mathbb{R}^{T_q \times C}, q \in [1, Q]\}$ where $Q$ is the total number of questions and $T_q$ is the embedding length of question $q$.

### 2.2.2 Memory Pointer Prompting

As processing all $N$ high-resolution visual embeddings with the LLM is computationally difficult, we propose to identify key visual embeddings in a question-aware manner and only send those selected embeddings to the subsequent LLM. Inspired by previous works on Pointer Networks (Vinyals et al., 2015; Merity et al., 2016), we propose a Memory Pointer Prompting mechanism, which is illustrated in Figure 4. Memory Pointer Prompting consists of two steps during inference: global glimpse and fallback. In the global glimpse step, key visual embeddings are identified from all frame-level embeddings, guided by the context of the question. During the subsequent fallback step, the important visual embeddings are selected, and their higher-resolution versions are provided to the LLM transformer backbone for further processing and language response generation.

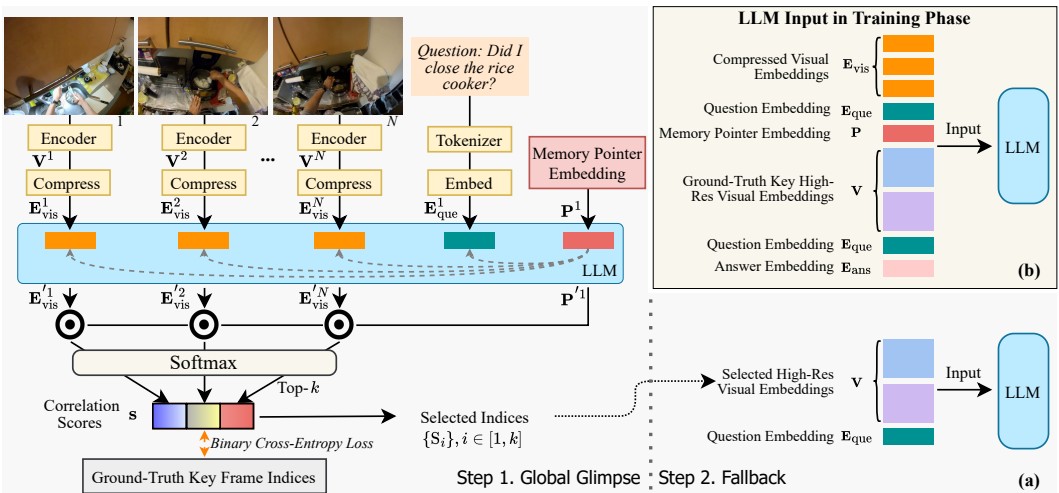

Figure 4: **(a)** Overview of the proposed Memory Pointer Prompting mechanism. Its inference consists of two steps: (1) *Global Glimpse*: We concatenate the compressed visual embeddings from all frames, denoted as $\mathbf{E}_{\text{vis}}^i$ for $i \in [1, N]$, with the question embeddings $\mathbf{E}_{\text{que}}^1$ and the memory pointer embedding $\mathbf{P}^1$. This combined embedding sequence is then input into the LLM. From the last layer, we extract embeddings and compute the dot product between the memory pointer embedding and all compressed visual embeddings to generate the correlation scores. The indices of the frames with the top $k$ scores are selected. During training, the correlation scores are supervised by ground-truth key frame indices via a binary cross-entropy loss. (2) *Fallback*: The high-resolution visual embeddings corresponding to the selected indices are fed into the LLM along with the question embeddings for final processing and response generation. **(b)** Illustration of LLM input sequence during training.

**Global Glimpse Step.** We begin by compressing the visual embeddings through average pooling along the embedding length dimension, resulting in a set of compressed visual embeddings $\{\mathbf{E}_{\text{vis}}^i \in \mathbb{R}^{1 \times C}, i \in [1, N]\}$. Next, we introduce a learnable memory pointer prompt embedding $\mathbf{P} \in \mathbb{R}^{1 \times C}$, duplicate it $Q$ times, yielding $\{\mathbf{P}^i \in \mathbb{R}^{1 \times C}, i \in [1, Q]\}$, and concatenate the embeddings as follows:

$$[\mathbf{E}_{\text{vis}}^1, \mathbf{E}_{\text{vis}}^2, ..., \mathbf{E}_{\text{vis}}^N, \mathbf{E}_{\text{que}}^1, \mathbf{P}^1].$$

Here $Q = 1$ as MLLMs generate answers for only one question at a time. In this way, the question embedding is followed by a pointer embedding, which will be used to identify key visual embeddings with knowledge of the question embedding. The entire embedding sequence is then fed into the LLM, from which we obtain the output embedding sequence of the final layer:

$$[\mathbf{E}_{\text{vis}}^{'1}, \mathbf{E}_{\text{vis}}^{'2}, ..., \mathbf{E}_{\text{vis}}^{'N}, \mathbf{E}_{\text{que}}^{'1}, \mathbf{P}^{'1}].$$

We extract and stack the processed visual embeddings $\{\mathbf{E}_{\text{vis}}^{'i} \in \mathbb{R}^{1 \times C}, i \in [1, N]\}$ to obtain the matrix $\mathbf{E}_{\text{vis}} \in \mathbb{R}^{N \times C}$. We conduct a softmax dot product operation between $\mathbf{E}_{\text{vis}}$ and $\mathbf{P}^{'1}$:

$$\mathbf{s} = \text{Softmax}(\mathbf{E}_{\text{vis}} \cdot \mathbf{P}^{'1\mathsf{T}}) \in \mathbb{R}^N. \tag{1}$$

Here $\mathbf{s}$ is a correlation score vector indicating the correlation between the question and each frame.

**Balancing Exploration and Exploitation.** Our approach to selecting key visual embeddings parallels the principles of Bayesian Optimization (Frazier, 2018), where the objective function is expensive to evaluate. In such cases, it's important to balance exploration (sampling in areas where the uncertainty is high) and exploitation (sampling in areas where the surrogate model predicts high performance). However, relying solely on the aforementioned Memory Pointer Prompting may lead to overemphasizing certain areas of interest, potentially undermining the exploration process. To mitigate this issue, we introduce perturbations into the score distribution by incorporating a uniform sampling distribution. The probability vector of uniform sampling can be written as:

$$\mathbf{u}^i = \begin{cases} \alpha & \text{if } i \in \text{linspace}(0, N, k), \\ 0 & \text{otherwise}. \end{cases} \tag{2}$$

Here $\alpha$ is an explore-exploit balancing parameter to adjust the probability distribution. We overlap the probability vector of uniform sampling and score matrix $\mathbf{s}$:

$$\mathbf{s} \leftarrow \mathbf{s} + \mathbf{u}. \tag{3}$$

We then identify the top-$k$ indices as the set $\{\mathbf{S}_i, i \in [1, k]\}$. In this way, we find the key visual embeddings in a question-aware manner.

Table 1: Distribution of videos and QA samples with different lengths.

| Class | Short | | Medium | | | Long | | Sum |
|---|---|---|---|---|---|---|---|---|
| Minutes | 0.5-1 | 1-2 | 2-4 | 4-10 | 10-20 | 20-40 | 40-60 | - |
| Videos | 100 | 100 | 100 | 100 | 100 | 100 | 29 | 629 |
| QAs | 500 | 498 | 987 | 997 | 1715 | 1792 | 537 | 7026 |

Table 2: Distribution of correct options in MCQs.

| Option | A | B | C | D |
|---|---|---|---|---|
| Counts | 1776 | 1751 | 1770 | 1729 |

**Fallback Step.** During inference, as shown in Figure 4, with the set of indices $\{S_i, i \in [1, k]\}$ for the selected visual embeddings, we now assemble the LLM input sequence as follows:

$$[ \quad \underbrace{\mathbf{V}^{S_1}, \mathbf{V}^{S_2}, ..., \mathbf{V}^{S_k}}_{\text{Selected Top-}k \text{ Visual Embeedings}} \quad , \mathbf{E}_{\text{que}}^1].$$

As previously introduced, $\mathbf{V}^{S_1}, \mathbf{V}^{S_2}, ..., \mathbf{V}^{S_k}$ denote the selected top-$k$ high-resolution visual embeddings, which provide more visual details than the compressed visual embeddings. This new embedding sequence is fed into the LLM to generate the final language response. In summary, the proposed Memory Pointer Prompting approach allows us to consider the full scope of video information while filtering out redundant data in the LLM transformer, ensuring computational efficiency. The new input serves as the final input of the LLM to generate the language response given the visual and textual information.

**Training Procedure.** Given the novel design of MM-Ego, its training procedure is different from popular MLLMs (Liu et al., 2023). Specifically, let the answer embedding for question $q \in [1, Q]$ be denoted as $\mathbf{E}_{\text{ans}}^q$, then the input embedding sequence during the training process is represented as:

$$[ \quad \underbrace{\mathbf{E}_{\text{vis}}^1, \mathbf{E}_{\text{vis}}^2, ..., \mathbf{E}_{\text{vis}}^N}_{\text{Compressed Visual Embeddings}} \quad , \mathbf{E}_{\text{que}}^1, \mathbf{P}^1, ..., \mathbf{E}_{\text{que}}^Q, \mathbf{P}^Q, \quad \underbrace{\mathbf{V}^{S_1}, \mathbf{V}^{S_2}, ..., \mathbf{V}^{S_k}}_{\text{Selected High-Res Visual Embeddings}} \quad , \mathbf{E}_{\text{que}}^1, \mathbf{E}_{\text{ans}}^1, ..., \mathbf{E}_{\text{que}}^Q, \mathbf{E}_{\text{ans}}^Q].$$

We also provide a simplified illustration (where $Q = 1$) of the input embedding sequence structure during training in Figure 4. Here, we begin by inputting the compressed visual embeddings for all $N$ frames, followed by the question embedding and memory pointer embedding. Next, we integrate the $k$ selected high-resolution visual embeddings (based on the ground-truth key frame labels), and finally, incorporate both the question and answer embeddings. Once the input sequence is prepared as outlined above, we can train MM-Ego similarly to traditional large language models. The compressed visual embeddings, question embedding, and memory pointer embeddings used as prefixes do not contribute to the language cross-entropy loss.

When training on samples from our curated egocentric QA dataset where there are ground-truth key frame labels for each question, we compute the correlation score vector $\mathbf{s}$ in the global glimpse step, and supervise it using a binary cross-entropy loss. For training samples that lack ground-truth key frame labels, we omit the prefixes, which results in the traditional MLLM training process.

## 3 EXPERIMENTS

In the experiment section, we will first present a new egocentric video understanding benchmark, specifically designed to assess episodic memory capabilities. Following this, we will perform comprehensive experiments to evaluate MM-Ego, utilizing both the newly introduced benchmark and existing public benchmarks.

### 3.1 EGOMEMORIA BENCHMARK

To evaluate the performance of egocentric MLLMs, especially in terms of episodic memory ability, we propose a new benchmark called EgoMemoria. Specifically, we generate memory-related questions and answers from human-annotated narrations in the validation set of the Ego4D dataset. To ensure diversity,

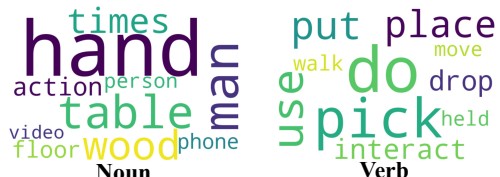

Figure 5: The most frequently occurring verbs and nouns in EgoMemoria.

for each video we only generate a limited number of questions. We divide the videos into seven different length ranges: 0.5 to 1 min, 1 to 2 min, 2 to 4 min, 4 to 10 min, 10 to 20 min, 20 to 40 min, and 40 to 60 min. We aim to balance the number of samples in different video lengths. The

**Question 1: Which hand did I use to touch the handle of the drilling machine?**
**Choices**: **A**: I touched the handle of the drilling machine with my right hand. **B**: I touched the handle of the drilling machine with both hands. **C**: I touched the handle of the drilling machine with my foot. **D**: I touched the handle of the drilling machine with my left hand.

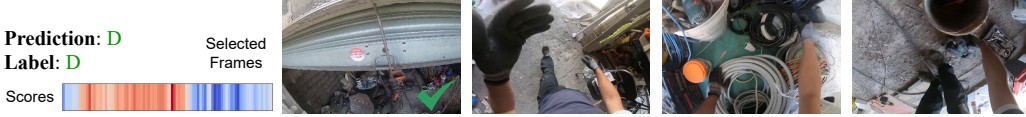

**Prediction**: D Selected Frames
**Label**: D
Scores

**Question 2: Did I use a ladder during the process?**
**Choices**: **A**: No, I used a step stool instead. **B**: No, I stood on a chair. **C**: Yes, I climbed down the ladder. **D**: No, I reached up without assistance.

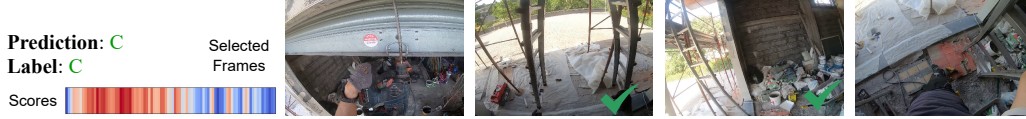

**Prediction**: C Selected Frames
**Label**: C
Scores

**Question 3: What did I connect to the socket?**
**Choices**: **A**: I connected an electric cleaner to the socket. **B**: I connected a digital clock to the socket. **C**: I connected a table lamp to the socket. **D**: I connected a coffee maker to the socket.

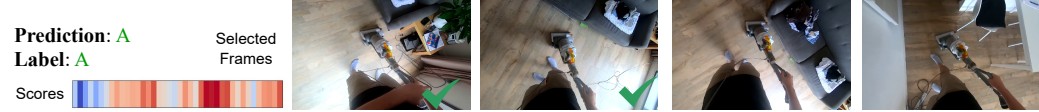

**Prediction**: A Selected Frames
**Label**: A
Scores

**Question 4: What color was the car that drove past last?**
**Choices**: **A**: 1. The car that drove past last was red. **B**: 2. The car that drove past last was blue. **C**: 3. The car that drove past last was black. **D**: 3. The car that drove past last was white.

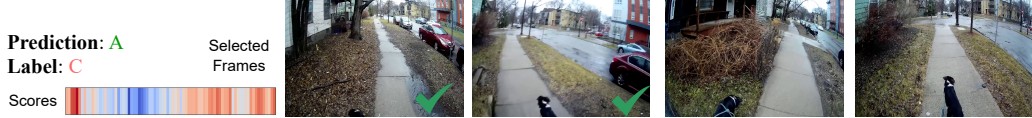

**Prediction**: A Selected Frames
**Label**: C
Scores

**Question 5: Where did I place the knife after peeling the zucchini?**
**Choices**: **A**: I left the knife in the sink. **B**: I put the knife back in the drawer. **C**: I placed the knife on the counter. **D**: I dropped the knife on the cutting board.

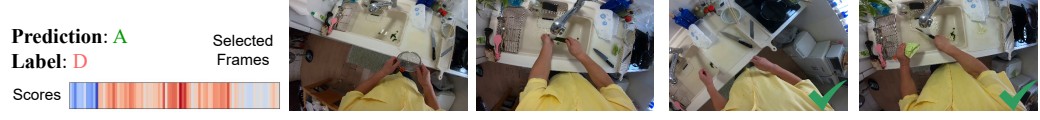

**Prediction**: A Selected Frames
**Label**: D
Scores

Figure 6: EgoMemoria QAs visualization and prediction analysis of the global glimpse step. We find high consistency between the identified key frames and the questions, demonstrating the effectiveness of the proposed Memory Pointer Prompting method. The visualized correlation scores show distinct distributions for different questions given the same video, indicating its question-specific nature. The ✓ indicates that the selected frames are relevant to the questions.

distribution of videos and corresponding question-answer pairs (QAs) for each category is shown in Table 1. Furthermore, we group these video lengths into three broader categories: short (0.5 to 2 min), medium (2 to 20 min), and long (20 to 60 min). In total, we collect 629 videos with 7,026 questions. The most frequently occurring verbs and nouns in the questions are visualized in Figure 5.

Since free-form answers are typically evaluated using a closed-source LLM as a judge, the evaluation can be inconsistent and subject to significant variance, especially due to model version updates. To ensure more reliable, standardized, and consistent performance evaluation, we convert the free-form answers into multiple-choice questions (MCQs), which helps reduce score instability. In practice, based on the free-form answer, we instruct ChatGPT to generate three additional choices that are plausible but incorrect, considering the original question and answer. We then randomize the order of these choices to achieve a uniform distribution of correct options, as shown in Table 2, to minimize bias in option placement. We visualize some randomly sampled examples in Figure 6.

## 3.2 EXPERIMENTAL SETUP

**Training Data.** We employ a joint image-video supervised fine-tuning (SFT) strategy. To enhance the model's capability in understanding a broader range of visual data, we combine our egocentric QA dataset with a variety of multimodal datasets. We curate an SFT dataset mixture consisting of our egocentric QA dataset, Ego4D narration dataset (Grauman et al., 2022), LLaVA-NeXT SFT

Table 3: Performance comparison and language bias analysis of different models on the EgoMemoria benchmark. Our MM-Ego model demonstrates the best performance both before and after excluding the language bias of different models.

| Method | LLaVA-OV (Li et al., 2024a) | | | | Ego SFT | | | | MM-Ego | | | |
|---|---|---|---|---|---|---|---|---|---|---|---|---|
| | Short | Medium | Long | Avg | Short | Medium | Long | Avg | Short | Medium | Long | Avg |
| Original | 70.24 | 64.94 | 61.19 | 65.45 | 79.06 | 76.34 | 73.51 | 76.30 | 79.96 | 79.64 | 79.09 | **79.56** |
| Exclude *LLaVA-OV* Bias | 56.44 | 49.64 | 44.83 | 50.30 | 66.37 | 64.15 | 60.03 | 63.52 | 71.97 | 70.68 | 68.15 | **70.26** |
| Exclude *Ego SFT* Bias | 55.75 | 49.27 | 45.21 | 50.08 | 61.73 | 59.59 | 54.50 | 58.61 | 67.70 | 66.33 | 63.89 | **65.97** |
| Exclude *MM-Ego* Bias | 47.41 | 42.11 | 35.22 | 41.58 | 50.60 | 46.39 | 40.38 | 45.79 | 49.80 | 49.11 | 43.81 | **47.58** |
| Mean Debiased Accuracy (MDA) | 53.20 | 47.01 | 41.76 | 47.32 | 59.56 | 56.71 | 51.64 | 55.97 | 63.16 | 62.04 | 58.62 | **61.27** |

collection (including ChartQA (Masry et al., 2022), AI2D (Hiippala et al., 2021), DocVQA (Mathew et al., 2021), DVQA (Kafle et al., 2018), COCO (Lin et al., 2014)), ShareGPT4V (Chen et al., 2023a), synthdog-en (Kim et al., 2021)), ShareGPT-4o (Chen et al., 2023b), ALLaVA instruct (Chen et al., 2024a), ShareGPT4Video (Chen et al., 2024b), sherlock (Hessel et al., 2022), ScienceQA (Lu et al., 2022), NExT-QA (Xiao et al., 2021), and ActivityNet-QA (Yu et al., 2019).

**Implementation Details.** The model is trained for 1 epoch with a base learning rate of $1 \times 10^{-5}$, using a cosine scheduler. The batch size is set to 128. We sample a maximum of 300 frames ($N = 300$) and select 32 visual embeddings in the proposed memory pointer prompting mechanism. By default, we set the explore-exploit balancing parameter $\alpha$ to 0.1. Greedy decoding is used in generation.

**Pretrained Models.** Our MM-Ego model is initialized from LLaVA-OV 7B (Li et al., 2024a), a state-of-the-art MLLM known for its good performance on general multimodal understanding tasks. Following the same architecture, we use the SigLip-so400M ViT (Zhai et al., 2023) as the visual encoder for embedding video frames and Qwen2-7B (Yang et al., 2024) as the LLM architecture.

### 3.3 MAIN RESULTS

We first conduct experiments on our EgoMemoria benchmark, primarily comparing three models: LLaVA-OV (Li et al., 2024a), its fine-tuned version using our MM-Ego SFT data mixture (referred to as "Ego SFT"), and our MM-Ego model, which incorporates the proposed Memory Pointer Prompting mentioned in Section 2.2.2. We show the EgoMemoria accuracy in the first row of Table 3. We observe a significant improvement in the model's performance on egocentric QAs after training on our MM-Ego data mixture, attributed to the rich egocentric knowledge provided by our curated egocentric QA training data. Moreover, leveraging the MM-Ego model architecture further enhances performance, thanks to the effective Memory Pointer Prompting mechanism.

However, we notice that the original overall performance metrics are higher than anticipated, raising curiosity about the extent to which language bias contributes to the models' accuracy. To answer this question, we conduct additional experiments aimed at eliminating these language biases. Specifically, we test the three model variants on the EgoMemoria benchmark without any visual inputs, identifying questions that could be correctly answered without videos as "language-biased questions". Then, we evaluate the models' performance on the subset of the benchmark without language-biased questions. For fairness, we apply this debiasing process across all three models so that they are evaluated on the same sets of data. We calculate the mean accuracy of the debiased variants, referred to as the "Mean Debiased Accuracy (MDA)". The results are presented in Table 3.

As expected, after removing the language-biased questions, the accuracy of all three models drops significantly to a more reasonable level. The performance decline is notably more pronounced in the "Medium" and "Long" classes compared to the "Short" class. For example, the average accuracy of LLaVA-OV across the three classes (short, medium, and long) drops from 65.45 to 47.32. The decrease in the "Short" class is 17.04, in the "Medium" class is 17.93, and in the "Long" class is 19.43. Despite this, we still observe improvements in MDA after training with SFT data generated by our MM-Ego data engine (**+8.65**) and applying our Memory Pointer Prompting method (**+13.95**). These results demonstrate the effectiveness of our approach even after considering language bias.

To better understand the capability of MM-Ego, we compare its performance with state-of-the-art video MLLMs on EgoMemoria and prevalent large-scale video QA benchmarks, including the long egocentric video understanding benchmark EgoSchema (Mangalam et al., 2023) and the Internet-video-based long-video understanding benchmark Video-MME (Fu et al., 2024). The results are shown in Table 4. On EgoMemoria, GPT-4o is evaluated using 32 uniformly sampled frames from the videos, while other models follow their respective official inference settings. The MDA on

Table 4: Comparison with state-of-the-art video MLLMs. MM-Ego shows strong performance on egocentric understanding and competitive performance on Internet video understanding.

| Method | EgoMemoria (MDA) | | | | EgoSchema | Video-MME (w/o subs) | | | |
|---|---|---|---|---|---|---|---|---|---|
| | Short | Medium | Long | Avg | Full | Short | Medium | Long | Entire |
| GPT-4o | **64.31** | 59.47 | 57.65 | 60.48 | **72.2** | **80.00** | **70.30** | **65.30** | **71.90** |
| LLaVA-NeXT-Video-7B-DPO (Zhang et al., 2024b) | 30.38 | 25.95 | 21.49 | 25.94 | - | - | - | - | - |
| LLaVA-NeXT-Video-32B-Qwen (Zhang et al., 2024b) | 43.78 | 33.76 | 31.04 | 36.19 | 60.85 | - | - | - | 60.20 |
| LLaVA-OV 7B (Li et al., 2024a) | 53.20 | 47.01 | 41.76 | 47.32 | 60.10 | 69.30 | 56.00 | 49.40 | 58.30 |
| MM-Ego (ours) | 63.16 | **62.04** | **58.62** | **61.27** | 69.03 | 67.60 | 55.70 | 47.80 | 57.00 |

Table 5: MDA on EgoMemoria when inferring with different numbers of frames. Our MM-Ego model shows a smaller relative drop on average when decreasing the number of sampled frames.

| Frames | Short | | | Medium | | | Long | | | Avg | | |
|---|---|---|---|---|---|---|---|---|---|---|---|---|
| | LLaVA-OV | Ego SFT | MM-Ego | LLaVA-OV | Ego SFT | MM-Ego | LLaVA-OV | Ego SFT | MM-Ego | LLaVA-OV | Ego SFT | MM-Ego |
| **32** | 53.20 | 59.56 | 63.16 | 47.01 | 56.71 | 62.04 | 41.76 | 51.64 | 58.62 | 47.32 | 55.97 | 61.27 |
| **16** | 52.68 | 60.45 | 63.82 | 46.37 | 55.99 | 60.81 | 40.12 | 51.15 | 58.16 | 46.39 | 55.86 | 60.93 |
| **8** | 50.76 | 59.59 | 62.22 | 44.82 | 54.55 | 58.23 | 39.41 | 49.11 | 55.19 | 44.99 | 54.42 | 58.55 |
| **4** | 50.43 | 55.36 | 62.30 | 42.54 | 52.08 | 58.44 | 38.88 | 48.40 | 54.65 | 43.95 | 51.95 | 58.46 |
| **Rel. Diff** | 5.20% | 7.07% | **1.36%** | 9.49% | 8.16% | **5.81%** | 6.89% | **6.26%** | 6.77% | 7.12% | 7.19% | **4.59%** |

EgoMemoria is computed using the debiased subsets used in Table 3. Notably, MM-Ego exhibits the highest performance on EgoMemoria, particularly in the 'Medium' and 'Long' classes. On the EgoSchema benchmark, our model achieves a substantial performance gain of **+8.18** over the previous state-of-the-art open-source model ("LLaVA-NeXT-Video-32B-Qwen"), underscoring the effectiveness of both our data and model design for egocentric understanding. Additionally, on the challenging Internet video understanding Video-MME benchmark, our model is on par with the leading model of similar parameter size although our data mixture is less diverse compared with (Li et al., 2024a). These results showcase MM-Ego's capability in egocentric video understanding while preserving its general video comprehension abilities.

## 3.4 MODEL ANALYSIS

**Quantitative Analysis of Different Numbers of Frames.** To evaluate the influence of sampling different numbers of frames for different models, we calculate the mean debiased accuracy (MDA) in Table 5. The relative performance drop from sampling 32 frames to sampling 4 frames is also calculated. As expected, all models exhibit a decrease in performance with fewer sampled frames. Notably, MM-Ego exhibits a smaller average performance drop when the number of frames is reduced due to its ability to identify key frames given lower computational budget. The relative performance drop in the short category is considerably smaller compared to the medium and long categories, likely because shorter videos require fewer frames to comprehend.

Figure 7: MDA scores with different $\alpha$ values for explore-and-exploit balancing.

**Qualitative Analysis of Memory Pointer Prompting.** In Figure 6, we present a qualitative analysis of the accuracy of Memory Pointer Prompting on EgoMemoria. We randomly select samples and visualize the key frames identified by the global glimpse step in Memory Pointer Prompting. The results show a strong alignment between the questions and the selected frames. In failure cases, we observe that the issues are often due to the ambiguity of the questions, causing the model to struggle with accurately localizing the key visual embeddings. Furthermore, the visualized correlation scores during the global glimpse step show distinct patterns across various videos and questions, confirming its effectiveness in selecting key visual embeddings tailored to the specific questions.

**Quantitative Analysis of Explore-Exploit Balancing Parameter $\alpha$.** As discussed in Section 2.2.2, we design an explore-exploit balancing parameter $\alpha$ to fuse the uniform distribution and the sampling probability computed by Memory Pointer Prompting. We illustrate MM-Ego's performance with varying values of $\alpha$ in Figure 7. The results show that $\alpha = 0.1$ achieves the best performance, while larger or smaller values of $\alpha$ tend to either over-explore or over-exploit.

**Conversation Examples by MM-Ego.** In Figure 8, we show a real-world demo of MM-Ego, where the input video is a 2-minute long egocentric video captured by a camera on an off-the-shelf wearable

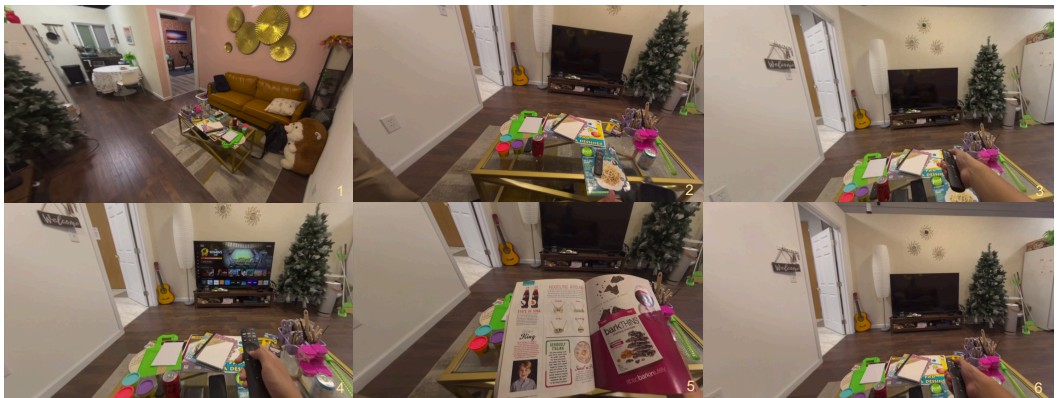

**Question 1: Where did I leave my wallet?**
**MM-Ego:** I left the wallet on the table. (Correct!)
**Question 2: Did I see anything I can play with?**
**MM-Ego:** Yes I have a guitar at the corner. (Correct!)
**Question 3: How many times did I interact with the remoter?**
**MM-Ego:** I interacted with the remoter three times. (Correct!)

*2 minutes video captured by a wearable device*

Figure 8: Real-world conversation examples generated by MM-Ego. The input is a 2-minute long egocentric video recorded using a camera on an off-the-shelf wearable device. MM-Ego can accurately identify key visual details and provide correct answers to the user's memory-related questions.

device (this video is not used in our dataset). MM-Ego is able to correctly answer the episodic memory-related questions given the egocentric video, despite the difference in data domain.

## 4 RELATED WORK

**Multimodal Large Language Models.** Recent advancements in Large Language Models (OpenAI, 2023; Touvron et al., 2023) have sparked significant interest in developing Multimodal Large Language Models (MLLMs) that combine the language understanding capabilities of LLMs with multi-modal perception abilities (Alayrac et al., 2022; Dai et al., 2023; Zhu et al., 2023; McKinzie et al., 2024). For video-based MLLMs, most works follow a structure akin to image-based MLLMs. To handle the large volume of video frames, some methods reduce the number of frames (Zhang et al., 2023a; Wang et al., 2024b; Maaz et al., 2024; Xu et al., 2024), which results in the loss of many visual details. Others extend the LLMs' context length by employing parallel techniques (Xue et al., 2024), but this often leads to low training efficiency. Unlike these approaches, our method preserves global awareness of the entire video, allows for attention to visual details, and is efficiently trainable.

**Egocentric Video Understanding.** While the growing field of egocentric video understanding is still in its infancy, there have been many influential works. For a comprehensive overview of egocentric vision please refer to Plizzari et al. (2024). On the data/benchmark side, representative works include Ego4D (Grauman et al., 2022), Ego-Exo4D (Grauman et al., 2024), and EPIC-KITCHENS-100 (Damen et al., 2018). When also considering language, prior work on egocentric video-language benchmarks include QaEgo4D (Bärmann & Waibel, 2022) and EgoSchema (Mangalam et al., 2023). For understanding long egocentric videos, prior modeling efforts include GroundVQA (Di & Xie, 2024), Encode-Store-Retrieve (Shen et al., 2023), and R-VLM (Xu et al., 2023). However, most previous works focus on classic video understanding tasks such as activity recognition and temporal grounding, and hence they do not involve a large language model for complex understanding. In contrast, we propose to develop an MLLM to tackle comprehensive egocentric video understanding.

## 5 CONCLUSION

In this paper, we make three key contributions towards the development of egocentric foundation models: the creation of a large-scale egocentric QA training dataset, the introduction of a novel model designed for effective long egocentric video comprehension, and the establishment of the EgoMemoria benchmark for assessing models' ability to capture visual details from egocentric videos. We hope that these efforts will benefit further research on egocentric MLLMs.

ETHICS STATEMENT

Our proposed method does not involve the creation or introduction of any new video content. All generated data is derived from publicly available, privacy-protected datasets (Grauman et al., 2022). The data is intended exclusively for academic research purposes and will not be used for any commercial applications. We have adhered to ethical standards by ensuring that no private or sensitive data has been used or compromised.

REPRODUCIBILITY STATEMENT

We provide a detailed explanation of the data synthesis process in our data engine in Section 2.1. We also elaborate on our model design in Section 2.2.2. Additionally, we outline the implementation details, including the training hyperparameters in Section 3.2.

ACKNOWLEDGMENT

This research is supported in part by the Early Career Scheme of the Research Grants Council (RGC) of the Hong Kong SAR under grant No. 26202321 and SAIL Research Project.

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

## A    MORE ANALYSIS OF MEMORY POINTER PROMPTING

To further assess the effectiveness of MM-Ego and the proposed Memory Pointer Prompting mechanism, we present additional visual results of key frame identification during the global glimpse step in Figure 9. MM-Ego demonstrates the capability to extract relevant visual information from a large set of frames based on the given questions.

**Question 1: Where did I walk towards with the hose in my hands?**
Choices: **A**: I walked backward towards a stone wall. **B**: I walked sideways towards a wooden shed. **C**: I walked around towards a brick pathway. **D**: I walked forward towards an iron fence.

**Prediction**: D        Selected
**Label**: D             Frames

**Question 2: What was the color of the tape I tried to remove from the wood?**
Choices: **A**: The color of the tape was blue. **B**: The color of the tape was red. **C**: The color of the my was green. **D**: The color of the tape was yellow.

**Prediction**: D        Selected
**Label**: D             Frames

**Question 3: Which hand did I use to pick the frying pan from the boot of the pickup truck?**
Choices: **A**: I picked a frying pan from the boot of the pickup truck with my right hand. **B**: 1. I picked a frying pan from the boot of the pickup truck with my left hand. **C**: 2. I picked a fryingpan from the boot of the pickup truck with both hands. **D**: 3. I picked a frying pan from the boot of the pickup truck using a cloth in my left hand.

**Prediction**: A        Selected
**Label**: A             Frames

**Question 4: What did I pass to my left hand?**
Choices: **A**: I passed the cup to my left hand. **B**: I passed the plate to my left hand. **C**: I passed the book to my left hand. **D**: I passed the remote to my left hand.

**Prediction**: B        Selected
**Label**: B             Frames

**Question 5: What action did I take with the frying pan at the end?**
Choices: **A**: A: I moved the frying pan on the cooker with my left hand and then stirred the content with the chopsticks. **B**: I placed the frying pan in the sink and washed it using a sponge and dish soap. **C**: I transferred the frying pan to the dining table and served the food onto plates. **D**: I hung the frying pan on the wall rack and wiped the stove clean.

**Prediction**: A        Selected
**Label**: A             Frames

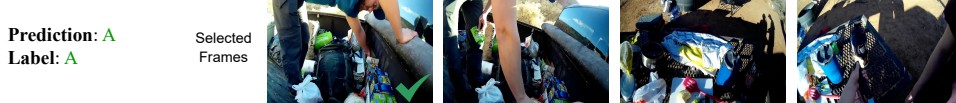

**Question 6: How did I add spice to the frying pan the first time?**
Choices: **A**: I grabbed a handful of spice and sprinkled it over the frying pan. **B**: I shook the spice container directly above the frying pan. **C**: I measured the spice with a teaspoon and added it to the frying pan. **D**: I scooped out some spice with the spoon and poured it in the frying pan.

**Prediction**: D        Selected
**Label**: D             Frames

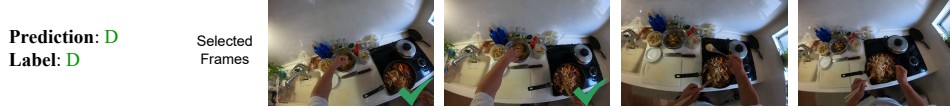

Figure 9: More key frame identification results of the global glimpse step on EgoMemoria. We find high relevance between the identified key frames and the questions, demonstrating the effectiveness of the proposed Memory Pointer Prompting method. The ✓ indicates that the selected frames are relevant to the questions.

## B  MORE DISCUSSION ON RELATED WORKS

**Egocentric QA Data Generation** After finishing the project, we find that the generation process in MM-Ego data engine shares some similar processes with the recently published LLaMA-VID (Li et al., 2024b) and GroundVQA (Di & Xie, 2024). LLaMA-VID utilizes movie synopses to prompt LLMs to produce movie summaries and plot-related QA pairs. GroundVQA generates short-term (around 8 minutes) episodic memory QA from video narrations, but the goal and implementation details are different. MM-Ego collects and processes videos with significantly more diverse video lengths from 30 seconds to 1 hour. The scale of our produced egocentric QA dataset is also significantly larger (7M vs. 303K).

**Long Video Understanding and Egocentric Understanding Evaluation Benchmarks** In recent years, there have been some pioneering benchmarks for assessing the performance of multimodal models in understanding long videos (Lei et al., 2018; Song et al., 2023; Zhang et al., 2023b; Fu et al., 2024; Wang et al., 2024a; Wu et al., 2024). Since the content of egocentric scenes differs from that of YouTube videos or movies, researchers have proposed specialized datasets for egocentric scene understanding. There are benchmarks for egocentric images (Cheng et al., 2024) and videos (Mangalam et al., 2023; Di & Xie, 2024; Bärmann & Waibel, 2022). QAEgo4D (Bärmann & Waibel, 2022) benchmark assesses shorter-term video understanding (around 8 minutes) with a considerably smaller dataset (1,850 questions across 166 videos). EgoSchema (Mangalam et al., 2023) has more video clips, yet the video lengths are still relatively short. Concurrent with our work, HourVideo (Chandrasegaran et al., 2024) introduces an important egocentric QA benchmark consisting of 121,976 QA samples across 500 videos, ranging in length from 20 to 120 minutes. Our EgoMemoria benchmark provides 7,026 QA samples for 629 videos and encompasses a wide range of video lengths, spanning from 30 seconds to 1 hour. The diverse video lengths make the benchmark significantly more challenging and closer to real-world egocentric video use cases. Furthermore, we contribute a large-scale egocentric QA training dataset with more than 7 million QA samples for 8,933 egocentric videos, which enables further research on training more powerful egocentric video understanding models. We show the statistics of these datasets in Table 6. Even when compared with other general/movie long video understanding benchmarks, the total numbers of QA samples and video counts in our Egomemoria benchmark are still significant.

| Benchmark | Videos | QAs | Video Length Distribution | Data Type |
|---|---|---|---|---|
| TVQA-test (Lei et al., 2018) | 1,089 | 7,623 | 0 - 3 min | TV shows |
| MovieChat-1K-test (Song et al., 2023) | 100 | 1,950 | 6 - 10 min | Movie |
| MoVQA (Zhang et al., 2023b) | 20 | 21,953 | 7.5 - 120 min | Movie |
| Video-MME (Fu et al., 2024) | 900 | 2,700 | 0 - 60 min | Internet |
| LVBench (Wang et al., 2024a) | 103 | 1,549 | 30 - 140 min | Internet |
| LongVideoBench (Wu et al., 2024) | 3,763 | 6,678 | 0 - 60 min | Mixed |
| EgoSchema (Mangalam et al., 2023) | 5,063 | 5,063 | 0.5 - 3 min | Egocentric |
| QAEgo4D-test (Bärmann & Waibel, 2022) | 166 | 1,850 | 0 - 8 min | Egocentric |
| GroundVQA-test (QAEgo4D_close) (Di & Xie, 2024) | 148 | 500 | 0 - 8 min | Egocentric |
| HourVideo (Concurrent Work) (Chandrasegaran et al., 2024) | 500 | 12,976 | 20 - 120 min | Egocentric |
| MM-Ego Training | 8,933 | 7M | 0.5 - 60 min | Egocentric |
| MM-Ego Evaluation (EgoMemoria Benchmark) | 629 | 7,026 | 0.5 - 60 min | Egocentric |

Table 6: Comparison with exiting long video understanding datasets.

**Limitation and Future Work** While MM-Ego demonstrates a strong ability in egocentric understanding, there is still room for further improvement. On the data and benchmark side, we can introduce more diverse egocentric understanding corpus (Grauman et al., 2024; Huang et al., 2024). For the model itself, we plan to enhance its capacity to process a larger number of frames, such as at the order of thousands, to better handle longer or even always-on egocentric videos.

## C  FINE-TUNING EXCLUSIVELY ON EGOCENTRIC QA DATA

To evaluate model performance when fine-tuning exclusively on egocentric QA data, we conduct an ablation study, with the results presented in Table 7. For both LLaVA-OV (Li et al., 2024a) and MM-Ego, we consider two variants: one fine-tuned on our comprehensive data mixture and the

other trained solely on egocentric QA data. The results demonstrate further performance improvements on the EgoMemoria benchmark. However, it is important to note that such domain-specific fine-tuning largely restricts the models' capacity for general video understanding. On the other hand, we observe that MM-Ego still achieves significantly better performance in egocentric video understanding, attributed to the learning of the memory pointer prompting mechanism.

| Name | Short | Medium | Long | Avg |
|---|---|---|---|---|
| LLaVA-OV (Data Mixture) | 59.56 | 56.71 | 51.64 | 55.97 |
| LLaVA-OV (Egocentric Data Only) | 62.58 | 58.13 | 53.61 | 58.11 |
| MM-Ego (Data Mixture) | 63.16 | 62.04 | 58.62 | 61.27 |
| MM-Ego (Egocentric Data Only) | 65.41 | 64.89 | 61.05 | 63.78 |

Table 7: Performance comparison of different models on EgoMemoria (MDA) with fine-tuning exclusively on egocentric QA data.

## D  ANALYSIS OF USING LANGUAGE MODEL IN DATA ENGINE

The motivation for using a language model to convert egocentric video captions into egocentric QA conversations is to address the "chicken-or-egg dilemma". If we rely on a vision-language model (VLM) to generate egocentric QA pairs, the quality of the data is inherently limited by the egocentric understanding capabilities of the labeling VLM. Consequently, downstream VLMs trained on this synthetic data cannot outperform the labeling VLM. This creates the chicken-or-egg problem: do we have a strong egocentric VLM first, or do we have good egocentric QA data first?

To address this dilemma, our "narration-to-egocentric QA" data engine leverages a language-only model to generate QA samples. This approach circumvents the hallucination and inaccuracy issues often associated with long video understanding, which is a very challenging task by itself. Since we have access to high-quality, fine-grained, and densely annotated video narrations created by human labelers in Ego4D, the essential visual information has already been effectively translated into the narrations.

We conduct a preliminary experiment of using image frames augmented GPT-4o to produce egocentric QA. We uniformly sample 32 frames with a resolution of 336x448 from a 30 seconds egocentric video.

**Input Narrations:** VID_NARRATION = ["I close the tap.", "I press the buttons on the rice cooker.", "I open the rice cooker.", "I stir the rice.", "I close the rice cooker.", "I press the buttons on the rice cooker."]

**Text Prompts:**
```
``Please design at most 5 questions and answers about
self-questioning my memory based on a video.  For example, ask
questions about what I have done, what objects did I interact,
where did I go, what did other people do, the locations, colors,
and shapes of the objects in the scenes if mentioned, based solely
on the video narrations $VID_NARRATION. The answers should be in
first-person perspective or objective description of the objects.
Remember this rule:  special word 'I', 'c', or 'C' represent me
(first-person perspective), special word 'x', 'y', 'Y', 'Z',
'z', 'o', 'O', 'X', etc.  represent somebody else, this is very
important.  Ask very diverse questions that you are sure about
the answers, and the questions cannot be answered without watching
the video, which means no common sense questions.  Don't generate
anything if you do not have enough information.  Attach index
of the sentence that answer is based upon.  Directly reply by
strictly following this example:  'QUESTION: Where did I leave
my key?  ANSWER:I put it on the table._0,2 QUESTION:Did I close
the door when I left?  ANSWER:Yes I did._3 QUESTION:What is the
color of the cup?  ANSWER:White._1''
```

**Generated Egocentric QAs:** We present the generated QAs by GPT-4o both with and without video input in Table 8 and Table 9. In this example, we observe that the generated QAs have similar quality whether videos are included or not. However, incorporating video into GPT-4o's input significantly increases the time and computational costs. Moreover, when applying VLM to a large volume of videos, it becomes challenging to verify if any visual hallucinations arise from the video understanding process. Therefore, we choose to use a mature pure language model in our data engine.

| Question | Answer |
|---|---|
| What did I do to the tap in the video? | I closed the tap. |
| How did I interact with the rice cooker buttons? | I pressed the buttons on the rice cooker twice. |
| What did I use to stir the rice? | I used a spoon to stir the rice. |
| Did I leave the rice cooker open after stirring the rice? | No, I closed the rice cooker. |
| At what points in the video was the rice cooker interacted with? | The rice cooker was opened, stirred, and closed. |

Table 8: GPT-4o generated egocentric QAs with video input.

| Question | Answer |
|---|---|
| What action did I perform on the tap? | I closed it. |
| What did I do after pressing the buttons on the rice cooker the first time? | I opened the rice cooker. |
| What did I use to stir the rice? | I stirred the rice. |
| How many times did I press the buttons on the rice cooker? | Twice. |
| What sequence of actions involved the rice cooker? | I pressed the buttons, opened it, stirred the rice, closed it, and pressed the buttons again. |

Table 9: Generated egocentric QAs without video input.

## E   MM-EGO'S CONNECTION WITH COGNITIVE NEUROSCIENCE

Our proposed "Memory Pointer Prompting" two-step mechanism is inspired by the way human naturally process videos. When answering a specific question about a long video, we typically start by quickly scanning the entire video to identify frames relevant to the question, which is similar to our "global glimpse" step. We then closely examine those frames to find the answers, which resembles our "fallback" step.

In the study of human visual perception system, researchers identify two distinct pathways for processing visual information in the brain: the magnocellular and parvocellular pathways (Zeki, 2015; Hochstein & Ahissar, 2002). Our "global glimpse" step mirrors the functionality of the magnocellular pathway which is responsible for handling information about large, fast-moving objects. On the other hand, our "fallback" step aligns with the role of the parvocellular pathway which specializes in processing details of small, slow-moving objects.

## F   FUTURE DIRECTION ON PROCESSING LONGER VIDEOS

To further enhance MM-Ego's capability to handle even longer videos, we can adopt two strategies. First, leveraging aggressive parallelism techniques, such as sequential and tensor parallelism (Xue

et al., 2024), can significantly extend the context length of the transformer model. This will extend the model's ability to do reasoning in more frames. Second, we can introduce a hierarchical structure to the compressed visual embeddings by further consolidating embeddings from multiple frames into a single representation. Then we can design multiple global glimpse steps, enabling the model to identify relevant frames in a coarse-to-fine manner.

