# OpenReview forum: "MMEgo: Towards Building Egocentric Multimodal LLMs for Video QA"
_ICLR.cc/2025/Conference — ICLR 2025 Poster_

### Official Review · Reviewer_g472 · 2024-10-21

**Soundness:** 2
**Presentation:** 3
**Contribution:** 3
**Rating:** 6
**Confidence:** 4

**Summary:**

The contributions of this paper include three aspects. 1. It implemented an automatic annotation process, providing a high-quality, large-scale QA dataset for egocentric videos. 2. It constructed a challenging egocentric QA benchmark, consisting of 629 videos and 7026 questions and introduced a new metric to mitigate inevitable language biases in evaluated models. 3. It proposed a novel model structure, including the global glimpse step and fallback step. By fine-tuning, MM Ego was built and demonstrated excellent performance in egocentric video understanding.

**Strengths:**

- A new model structure is proposed, with the motivation of incrementally understanding videos: first, providing an overview of the entire video, then focusing on specific details, and keeping in mind particular questions.

- This paper provides dialogue examples of MM Ego in real-world scenarios, offering a paradigm for understanding human instructions in real-world settings.

- This work provides a large amount of Ego video data to the community through an automated annotation process and it is valuable to assess language biases in the LLM evaluation process.

**Weaknesses:**

- The authors suggest that in terms of benchmarking, existing egocentric video benchmarks either focus on shorter videos, such as EgoSchema and QaEgo4D or on internet video content, such as video MME. This results in a significant gap in egocentric video understanding benchmarks, but the data for the proposed benchmark EgoMemoria still comes from Ego4D. I do not feel that EgoMemoria is significantly different from the benchmarks mentioned earlier, and the authors need to clarify this point further.

- Is it reasonable to only use GPT-4o in the automated annotation process? When generating QA pairs, I think that videos are equally important as dense captions, which will further ensure the quality of the QA pairs. The authors need to provide ablation experiments to validate that the existing method is more reasonable and ensures higher annotation quality.

- To ensure reproducibility, the authors need to provide the prompt templates for GPT-4o and ChatGPT used in the automated annotation and evaluation processes.

- The benchmark proposed in this work includes 629 videos. The authors classified them by length but did not provide the distribution of videos for each length category. This is crucial for the comprehensiveness and robustness of model evaluation. The authors should further elaborate on the distribution of video lengths.

- In Table 4, MM Ego's performance on Video-MME is not as impressive as LLaVA OV. The authors should provide a more comprehensive analysis to explain the reasons for this discrepancy. Otherwise, I might conclude that the construction method of MM Ego sacrifices its ability to understand general videos in favor of enhancing ego understanding.

- Language bias is inevitable in LLM, but whether it will be influenced by random responses in ego videos also needs to be verified through an ablation study.

- The Global Glimpse Step and Fallback Step described by the author would be more complete and convincing if they could be correlated with mechanisms related to cognitive neuroscience and brain science.

- Figure 5 may be better represented using a word cloud.

- Please note that it is not GPT4-o, but GPT-4o.

If the author can address the questions above, I will improve my rating.

**Questions:**

Please refer to "Weaknesses".

---

### Official Review · Reviewer_JQHv · 2024-10-22

**Soundness:** 3
**Presentation:** 4
**Contribution:** 3
**Rating:** 6
**Confidence:** 4

**Summary:**

This paper introduces three aspects of a multimodal foundation model for egocentric video understanding. First, it proposes a data engine designed to generate egocentric QAs based on human-annotated narrations automatically. Specifically, it relies on a text-only language model (GPT4-o) to generate the QAs. Second, it presents a Memory Pointer Prompting method, which can help generate question-related responses by identifying key visual details. Third, it introduces a new benchmark called EgoMemoria and a new metric that effectively de-biases language data in the evaluation process.

**Strengths:**

- The new set of annotations introduced in the proposed dataset does look practical and timely, considering that it is based on narrations annotated by humans instead of language models. The dataset should be of higher quality than other synthetic datasets using large language models as pseudo-labelers.
- The proposed benchmark, EgoMemoria, with the suggested debiased metric, also looks interesting and well-defined. The QAs are distributed well for different video lengths, and the answers are uniformly distributed between multiple choices.
- The proposed MM-Ego model with the “Memory Pointer Prompting” mechanism is straightforward and intuitive. It understands a video in a progressive way, first getting an overview and then focusing on details.
- The paper is well-written and is easy to follow. It looks well-organized, with the required figures and tables placed throughout.

**Weaknesses:**

- [W1] The introduced data engine works only with human-annotated narrations. In other words, it is not trivial to scale up since it relies on human labor. The data engine looks like it is designed explicitly for the Ego4D dataset, which has human-annotated video clip narrations. I am not too sure if the data engine is a practical and valuable contribution.
- [W2] The authors have not compared the introduced EgoMemoria with other datasets. Even if it has high-quality data thanks to the use of human annotations, it is hard to say 7k MCQs large-scale. The number of annotations in the test split would be even smaller. I believe the authors should make a table comparing the EgoMemoria with other egocentric datasets regarding the number of clips, annotations (QA pairs), and clip lengths.
- [W3] The authors have not shown the results when using only the EgoMemoria for training since they use an SFT dataset mixture. It would also be better to include the results using only the EgoMemoria for training.

**Questions:**

What is the performance of LLaVa-OV and MM-Ego when using only the EgoMemoria for training?

---

### Official Review · Reviewer_bd41 · 2024-11-01

**Soundness:** 3
**Presentation:** 4
**Contribution:** 3
**Rating:** 6
**Confidence:** 4

**Summary:**

This work takes a step towards building egocentric MLLMs by contributing a large-scale dataset, a challenging benchmark, and an innovative model architecture. The model, named MM-Ego, demonstrates strong performance on egocentric video understanding. It is designed to process and understand long egocentric videos by progressively understanding the video: first getting an overview, then focusing on specific details with particular questions in mind.  The paper introduces also an egocentric QA benchmark called EgoMemoria, which contains 7,026 questions for 629 videos of different lengths. It also introduces a de-biasing evaluation method to mitigate language bias in model evaluations.

**Strengths:**

1. The authors curate a large-scale dataset with over 7 million QA samples for egocentric videos of varying lengths and introduce a QA benchmark called EgoMemoria, which contains 7,026 questions for 629 videos of different lengths.

2. A specialized multimodal architecture is proposed, using a "Memory Pointer Prompting" mechanism. This includes a global glimpse step for overarching video understanding and a fallback step that uses key visual information to generate responses, enabling the model to comprehend extended video content more effectively.

**Weaknesses:**

1. The first claim of contribution, the “narration to egocentric QA” data pipeline, I believe, should not be emphasized as a major contribution. This type of generating QA from dense captions have been used in multiple previous works, from the non-LLM era (like TVQA) to the LLM era (like LLama-VID). I believe it is better to tone down this statement.
2. The generated EgoMemoria Benchmark does not stands itself out of many long video understanding benchmarks. Even we narrow down to only egocentric videos, the GroundVQA dataset is also good to be compared and especially be used to test the MMEgo model. I would also recommend the authors to compare a long of long video datasets, providing more proofs that this benchmark is not so incremental.

Overall, I think this paper is proposing a good model, while the benchmark side is relatively weak. I would recommend the authors to either tone down the benchmark in the paper to put more emphasize on this model, or improve the benchmark. I recommend the authors to consider using the datasets of EgoExo4D and EgoExoLearn, both of which also contain dense narrations of egocentric videos and should be very suitable for enriching your benchmark in terms of both size and diversity.

**Questions:**

It would be great if the authors could answer my two points in the weakness section.

---

> ### Comment · Reviewer_bd41 · 2024-12-03
>
> Thanks to the authors for their detailed feedback. The authors nicely addressed my concerns. My final rating is still **weak accept** (higher than 6: marginally above the acceptance threshold but lower than 8: accept, good paper).
> I appreciate that the authors include comparisons with recent long-form video understanding benchmarks, and also showing a promising new model. The primary reason I have not given a higher rating is the lack of contribution on the benchmark side. I believe the key point for the benchmark to the community is its evaluation set, and the size of the instruction tuning set should not be emphasized and serve as a key contribution. Without a breakthrough contribution, giving a higher rating is hard for me.

---

### Official Review · Reviewer_hocP · 2024-11-06

**Soundness:** 3
**Presentation:** 3
**Contribution:** 3
**Rating:** 6
**Confidence:** 3

**Summary:**

The paper introduces MM-ego to process and understand long egocentric videos. It includes "narrations to QA" strategies for creating scalable training data. The paper also introduces a new benchmark called EgoMemoria to assess the ability of reasoning and memorizing visual details and evaluate the impact of language biases. The final contribution is the MM-Ego model which is based on a progressive memory pointing prompting consisting of global compressed features and fallback aka learnable memory pointers.

**Strengths:**

Pros:
MM-Ego data engine: augmenting Ego4D dataset with scalable QA is valuable
Ego-memoria benchmark is a good contribution, so is the MM-Ego model
the assessment of the impact of language bias is useful, and also shows the value of the data engine

**Weaknesses:**

The the model struggles with long videos. Whereas, egovideos are known for always ON camera meaning the ability to process long / unlimited length video is utmost important. It's a general research question for the community.

**Questions:**

What are your thoughts on enabling more number of frames into the reasoning pipeline?

---

### Meta-Review · Area_Chair_HKyQ · 2024-12-21

**Metareview:**

This paper proposes a new set of QA annotations for the Ego4D dataset, and a new multimodal architecture for video QA.

All the reviewers recognise the value in the dataset and think it will make for an interesting contribution to the community.  They especially appreciate having the human annotated data as part of the benchmark.

Some weaknesses pointed out by the reviewers include various experimental comparisons.  Others question the issue of over-claim certain contributions which already exist in the literature.  These concerns are mostly addressed, such that the final score is 4x borderline accepts.

The AC concurs with the reviewers that the dataset will make for a good contribution at ICLR.  However, the authors are requested to amend their manuscript to include their various clarifications and additional experimental comparisons for the camera ready.

The issue of over-claims on contribution and overall transparency is concerning.  Two specifics include
(1) the paper title, with the focus on multimodal LLM is broader than the scope of the contributions which is only in the form of QA.
(2) the abstract is vague in not stating the origins of the video data - i.e. Ego4D.  This should be clearly  stated so as not to give the impression that both the video data and annotations are new.

**Additional Comments On Reviewer Discussion:**

Some weaknesses pointed out by the reviewers include various experimental comparisons.  Others question the issue of over-claim certain contributions which already exist in the literature.  These concerns are mostly addressed, such that the final score is 4x borderline accepts.

---

### Decision · Program_Chairs · 2025-01-22

Accept (Poster)